# EPL-UFLSID: Efficient Pseudo Labels-Driven Underwater Forward-Looking Sonar Images Object Detection

Cheng Shen
Shanghai University
Shanghai, China
chauncey31@163.com

Liquan Shen*
Shanghai University
Shanghai, China
jsslq@163.com

Mengyao Li
Shanghai University
Shanghai, China
sdlmy@shu.edu.cn

Meng Yu
Shanghai University
Shanghai, China
ym19821214515@163.com

## Abstract

Sonar imaging is widely utilized in submarine and underwater detection missions. However, due to the complex underwater environment, sonar images suffer from complex distortions and noises, making detection models hard to extract clean high-level features for detection. Existing works introduce denoised images as pseudo labels to assist the network to extract clean features while not fully considering the rationality of pseudo labels. To this end, we propose an Efficient Pseudo Labels-Driven Underwater Forward-looking Sonar Images Object Detection algorithm (EPL-UFLSID). Specifically, we first design a Gaussian Mixture Model based Deep Image Prior (GMMDIP) network to generate denoised sonar images by setting the GMM distribution as its input. After that, to filter the most detection-friendly images of the denoised images generated by GMMDIP as efficient pseudo labels, Detection-Friendly Image Quality Assessment network (DFIQA), is designed, which is also able to help EPL-UFLSID further distill cleaner features from pseudo labels to improve detection performance. Extensive experimental results show that our EPL-UFLSID reaches average precision (AP) of 67.8%/39.8% and average recall (AR) of 73.7%/49.6% on two real sonar datasets, which outperforms SOTA underwater forward-looking sonar images object detection algorithms.

## CCS Concepts

• **Computing methodologies** → **Object detection**; *Reconstruction*.

## Keywords

Object Detection, Image Enhancement, Image Quality Assessment, Underwater Forward-Looking Sonar

*Corresponding author

**ACM Reference Format:**
Cheng Shen, Liquan Shen, Mengyao Li, and Meng Yu. 2024. EPL-UFLSID: Efficient Pseudo Labels-Driven Underwater Forward-Looking Sonar Images Object Detection. In *Proceedings of the 32nd ACM International Conference on Multimedia (MM '24), October 28-November 1, 2024, Melbourne, VIC, Australia.* ACM, New York, NY, USA, 9 pages. https://doi.org/10.1145/3664647.3681160

## 1 Introduction

Forward-looking sonar (FLS) is commonly employed as an equipment to gather underwater information due to its capbility to capture visible images in turbid and dark environments. Consequently, autonomous underwater vehicles (AUVs) are frequently equipped with FLS to aid in tasks such as positioning, object detection and other works [1][40]. In response to this need, numerous underwater FLS image object detection algorithms have been developed to enhance the efficiency and accuracy of AUV operations.

Generally, sonar images are easily affected by environmental noises, reverberant noises and self-noises [14], which heavily damage the quality of sonar images, resulting in bad detection performance. It's natural to think that when various noises are removed, AUVs can detect the object more accurately. Therefore there are a variety of methods including traditional and deep learning methods [39][20] developed to remove the noises from sonar images, while they simply model the sonar image noises like speckle noise or reverberant noise with a known and simple distribution.

In addition, although visually, the denoised sonar images look better for people to view and become smoother, they may not necessarily assist in the performance improvement of detection models when denoising the sonar images before feeding them into a detection model. This is because the purposes of denoising for view and denoising for detection are different, where exists potential conflicts between them [13]. In other words, the denoised sonar images may have noises that are not observable to the human eye, and accordingly reduce performance of subsequent detection models. A few works [18][7] have tried to handle the above problems by fusing features extracted from the denoised images and the original images that are favorable for detection, while their performance is still moderate due to the mediocre quality of the denoised images. Therefore, how to obtain higher quality denoised images and rationally introduce them into the detection backbone network as pseudo labels to enhance the detection performance still needs to be researched.

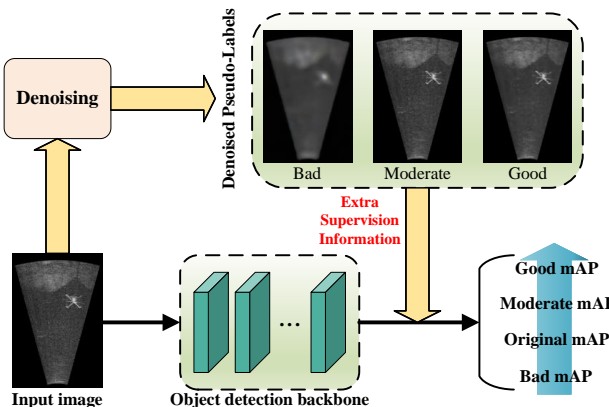

**Figure 1: The illustration of the enhancement of detection performance through the utilization of pseudo labels. Introducing denoised sonar images into the detection module can provide additional supervisory information to the detection backbone network, thereby enhancing the detection performance. However, the variability in the quality of denoised sonar images poses a challenge. When low-quality images are introduced into the detection module as pseudo labels, they may even degrade the detection performance. In other words, the detection performance are positively correlated with the quality of denoised sonar images which are selected as pseudo labels.**

To this end, we propose an Efficient Pseudo Labels-Driven Underwater Forward-looking Sonar Images Object Detection algorithm, namely EPL-UFLSID, which introduces efficient pseudo labels to drive detection models to extract cleaner features and achieve better detection performance. Firstly, although Deep Image Prior (DIP) [29] can achieve excellent performance in denoising natural images, its effectiveness diminishes when applied to gray, low-contrast, and noise-disturbed sonar images. We design a Gaussian Mixture Model (GMM) [24] based DIP network (GMMDIP) to generate higher quality denoised sonar images, which utilizes the Gaussian mixture model acquired by fitting the original sonar images as its input, thereby providing GMMDIP with a prior on the distribution of sonar images. After GMMDIP provides a batch of denoised images with varying denoising qualities, the denoised image of the highest quality need to be selected from them as a pseudo label based on the illustration in Figure 1. Therefore, we design a Detection-Friendly Image Quality Assessment network (DFIQA) to select the best one from a set of denoised sonar images generated from GMMDIP which will be served as a pseudo label to help improve the detection performance. Besides, DFIQA provides a score for the reconstructed image compared to pseudo label, which will be served as a loss after sigmoid function to jointly optimize the detection network. Our contributions can be summarized as follows:

**1)** We propose a novel algorithm named as EPL-UFLSID, which improves the forward-looking sonar images object detction performance by introducing efficient pseudo labels to guide the detection backbone network to extract cleaner detection-friendly features.

**2)** GMMDIP, an unsupervised sonar image denoising network, is proposed to produce high quality denoised sonar images which are served as pseudo labels to help improve the detection performance.

**3)** A detection-friendly image quality assessment network named as DFIQA is proposed to select the most efficient pseudo labels for detection, which is also a machine vision oriented image quality assessment network.

## 2 Related Work

### 2.1 Underwater Sonar Image Object Detection

In recent years, with the development of deep learning, the performance of CNN-based sonar image object detection algorithms has been greatly improved. For example, in addressing the challenge of improving underwater sonar image detection accuracy with finite training data, notable contributions have been made by studies such as [30], [31], and [32]. Among these, [30] stands out for achieving high accuracy through CNN network, surpassing the performance of certain template matching detection methods. [4] devises a feature extraction network that employs residual blocks to substitute the backbone in Mask RCNN, which reduces network parameters without compromising accuracy.

Besides, in order to reduce the complexity of the network as well as the running time, one-stage networks such as YOLOv3 [21], SSD [17], RetinaNet [15] are also adopted as the base detection network. [38] introduces an attention mechanism in YOLOv5 and proposes a real-time target detection algorithm TR-YOLOv5, which achieves some degree of improvement. [34] designs a multi-scale convolution structure to enhance the network perception of small target features and improves the performance of small target detection. [10] proposes a dual path feature fusion network for feature extraction, which achieves robust and real-time sonar image detection. However, these algorithms do not fully take into account the characteristics of the underwater sonar image such as grayscale, low contrast and diverse noise interference.

Recently, [19] denoises the sonar images by characterizing the noise of sonar images as multiplicative speckle noise, and sets them as pseudo labels to enhance the detection performance, however the quality of pseudo labels still needs to be improved. Therefore, how to generate better and efficient pseudo labels that provide additional supervisory information for the detection network to improve the sonar image detection performance still needs to be studied.

### 2.2 Underwater Sonar Image Denoising

The underwater sonar images captured by forward-looking sonar are subject to a variety of noise interference owing to its imaging mechanism and working environment, thus undermining image recognition, even detection tasks. Therefore, various traditional and deep learning methods for underwater sonar image denoising have been proposed.

Traditional methods usually need modeling specific noises based on a great amount of parameters and are less generalized. LEE filter [12], KUAN filter [11], FROST filter [5], and SRAD filter [37] are classical adaptive despeckling filters. [28] proposes a 2-D finite impulse response (FIR) Wiener filter driven by an adaptive cuckoo search (ACS) algorithm and eliminated Gaussian noise. However,

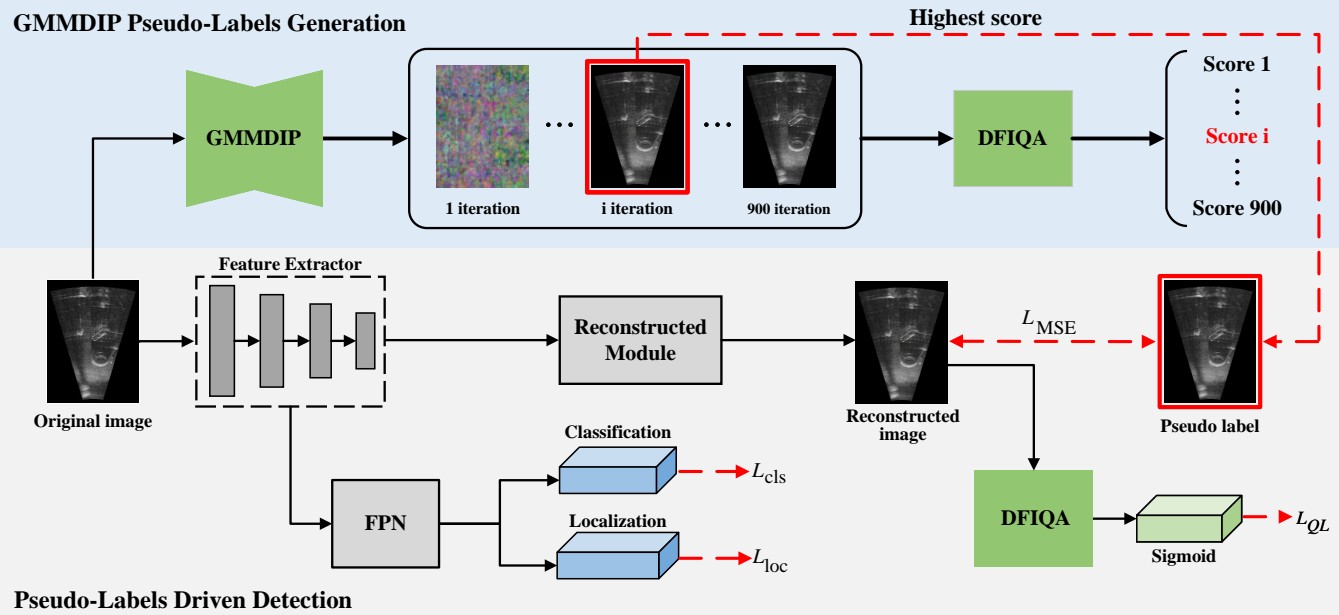

**Figure 2: Overview of the proposed method EPL-UFLSID. The original sonar images are denoised by GMMDIP, generating 900 denoised sonar images with various qualities. Then the denoised sonar image with the highest score is selected as an efficient pseudo label by DFIQA. Meanwhile, the original sonar image goes through a feature extractor along with a reconstruction module responsible for reconstructing a sonar image, which is a medium to allow the pseudo label and score generated by DFIQA on the reconstructed image to jointly optimize the entire network.**

the above traditional methods are generally ineffective because they damage the details and textures of the original sonar images during the denoising process.

Recently, there are many deep-leaning methods for underwater sonar image denoising. [9] uses the autoencoder algorithm for sonar image denoising. [35] proposes an image despeckling convolutional neural network (ID-CNN), which assumes a gamma distribution of noise. It uses a componentwise division residual layer to recover the speckle component. [2] proposes a self-supervised training strategy simplifying the generation of training sets for removing speckle noise from acoustic images. However, these methods are premised on the assumption that speckle noise is mainly presented on the sonar images, without considering complex noise distributions in real sonar images. Besides, due to the lack of noise-free sonar images, supervised methods require synthesizing the dataset with optical images, which introduce the domain shifting. As discussed in [41], the success of CNN-based denoisers is strongly dependent upon whether the distributions of noises in synthetic and realistic noisy images are well matched. Therefore, an unsupervised denosing algorithm for underwater sonar images without synthetic training datasets need to be studied.

## 3  Proposed methods

The architecture of the proposed method EPL-UFLSID is described in Figure 2. As mentioned before, the introduction of efficient pseudo labels can provide additional supervisory information for the detection network to extract cleaner feature information from the original sonar images. Therefore, we first design the GMMDIP

network, which generates a batch of sonar images with different qualities by fitting a Gaussian mixture model of the original sonar images as its input, giving in advance the distributional characteristics of the original sonar images for GMMDIP network. Then we design a siamese-like network with the help of entropy images named as DFIQA that selects the sonar image that is the most favorable for detection as an efficient pseudo label and DFIQA also provides a score for the reconstructed image, which will be served as a loss after sigmoid function to jointly optimize the detection network.

### 3.1  GMMDIP network

As previously explained, the denoised sonar images can be introduced as pseudo labels to provide additional supervisory information to the detection network. The quality of pseudo labels largely dedicates the effectiveness of the supplementary supervisory information extracted by the detection network, consequently affecting detection performance.

As DIP [29] has shown great potential in the field of image restoration, denoising, etc, and does not require a training dataset, we hence propose the Gaussian Mixture Model based Deep Image Prior network(GMMDIP) to generate a series of reliable denoised sonar images served as pseudo labels.

Here, we use $f_\theta$: $Z \rightarrow Y$ to denote the GMMDIP network parameterized by $\theta \in \Theta$, which transforms a tensor/vector $z \in Z$ to the original sonar image $y \in Y$. As stated in [29], the conditional image distribution $p(x^*|y_0)$ is modeled by using the prior knowledge of the network to recover the denoised image, where the unknown

**Figure 3: The structure of GMMDIP. The input to GMMDIP is obtained by fitting a Gaussian Mixture Model of the original image, corresponding to the fitted curve of the GMM distribution and the curve of the original image distribution. Meanwhile the shallow and last few convolutional layers in GMMDIP are frozen and used to enhance the convergence speed of the network.**

image $x^*$ is determined by the measurement $y_0$. Thus, the final optimization objective of GMMDIP can be interpreted as:

$$\theta^* = \arg\min_\theta E(f_\theta(z); y_0), \tag{1}$$

that is to say:

$$\min_\theta \|f_\theta(z) - y_0\|^2. \tag{2}$$

The structure of GMMDIP is shown in Figure 3, which is "hourglass" (also known as "encoder-decoder") like architecture with some skip connections. In previous research on DIP network optimization [6][27][22], few studies conducted on network input $z \in \mathbb{R}^{3 \times W \times H}$. Moreover, while the DIP network is commonly applied to address noises in natural images, the noise characteristics in sonar images are notably more intricate and challenging to simulate using conventional distributions. Additionally, sonar images contain less color information compared to natural images.

Therefore, we first propose to use the noise distribution obtained by fitting a Gaussian Mixture Model (GMM) to the original sonar image, namely $z = GMM(y_0)$, given the powerful ability of GMM to model complex and poorly defined distributions, instead of using the uniform noise between 0 and 0.1 as input. In this way, the initialized GMM input obtained for each sonar image can be provided to the network in advance with the initial distribution prior of the sonar image, which further improves the performance of denoising in conjunction with the depth prior learned by the GMMDIP network.

To further improve the denoising performance, we use the total variation (TV) [25]:

$$TV(\boldsymbol{x}) = \sum_{i,j} |\boldsymbol{x}_{i+1,j} - \boldsymbol{x}_{i,j}| + |\boldsymbol{x}_{i,j+1} - \boldsymbol{x}_{i,j}| \tag{3}$$

for any 2D image $\boldsymbol{x} \in \mathbb{R}^{h \times w}$. Thus the final optimized loss function for our GMMDIP network is:

$$\min_{\theta, \theta_{BN}} \|f_\theta(z) - y_0\|^2 + \lambda TV(f_\theta(z)), \tag{4}$$

where $\theta_{BN}$ means the parameter of batch normalization layer in GMMDIP, and $\lambda$ is a regularization parameter, set to 0.45.

## 3.2 DFIQA network

As mentioned before, the merit of the pseudo labels introduced in the detection network determines the ability of the detection backbone network to extract clean features. We therefore design the DFIQA network in the hope of accurately selecting one of the most detection-friendly images from a batch of denoised images to be detected as a pseudo label.

### 3.2.1 *DFIQA Dataset Constuction*.

Currently, image quality assessment algorithms primarily prioritize human eye perception, neglecting quality assessment methods for machine vision, especially in detection tasks. Differences in the properties of machine vision and human eye vision can lead to the fact that evaluation criteria specific to human eye vision are not applicable to machine vision tasks. Moreover, specific datasets for machine vision are scarce. Constructing datasets is crucial for training a quality evaluation network capable of generating detection-friendly labels. Therefore, a set of pseudo labels are firstly generated by GMMDIP for 900 iterations. Specifically, starting from iteration 100, one denoised sonar image is selected as a pseudo label every 100 intervals, resulting in 9 denoised sonar images for each sonar image. Besides one original image is also plused to serve as a pseudo label for each sonar image, thus 10 sets of pseudo labels are yielded to serve as quality assessment datasets. Furthermore, to enhance the generalization of the datasets, 10 sets of pseudo labels are applied to three one-stage detection networks, YOLOv3 [21], SSD [17] and RetinaNet [15]. Then, in order to evaluate these three detection networks, the product of two evaluation metrics commonly used in target detection, IoU and AP, is introduced, which is defined as follows:

$$MOS_{DFIQA} = AP \times IoU(\Omega_g, \Omega_p), \tag{5}$$

where $\Omega_g$ and $\Omega_p$ are the sample box area and the prediction box area, respectively. Besides, we weight and average the test results of the three different detection frameworks to obtain a more generalized MOS value. Finally, two different datasets are constructed with 18,680 images and 92000 images, representing the merits of the performance of detection models driven by different pseudo labels.

### 3.2.2 *Network Structure of DFIQA*.

After creating the DFIQA dataset, we develop an image quality assessment network to select optimal images for detection as pseudo labels from a series of denoised images with varying denoising qualities generated from GMMDIP. Additionally, the quality scores are integrated into the reconstruction part of EPL-UFLSID as a loss function, jointly optimizing the whole network.

The DFIQA network structure, as depicted in Figure 4, comprises two identical branches sharing weights, which select pairs of denoised images and their entropy images from DFIQA datasets as inputs. Given the positive correlation between the sharpness of images and high-level task performance like detection, DFIQA incorporates the entropy map as auxiliary information, which aids the feature extraction network in capturing features conducive to the detection task. In detail, the input image will go through a

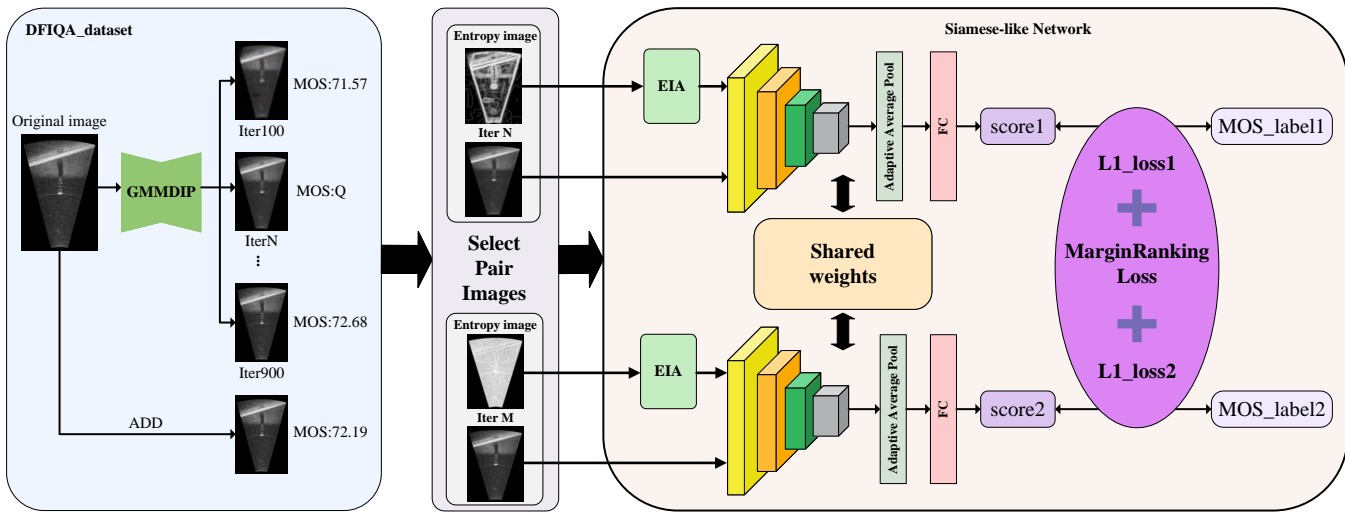

**Figure 4: The structure of DFIQA is a siamese-like network. In the training phase: We pick two images from DFIQA_Dataset as inputs and the entropy images of these two images will be used as auxiliary information to provide more features about the detection task for DFIQA's feature extraction network through the EIA module. The outputs of two branches are passed to the loss module, where we can compute the loss function and apply back-propagation to update parameters of the whole network. In the testing phase: We extract a single branch from the network to predict the image quality.**

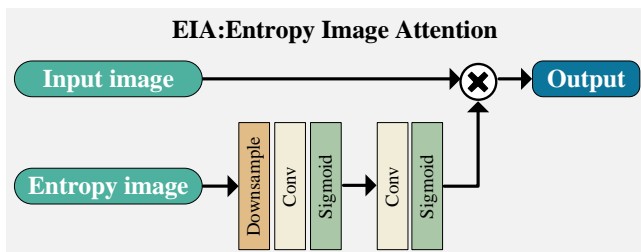

**Figure 5: The structure of EIA Module.**

ResNet50 feature extractor, while the entropy image of the input image will go through entropy image attention module (EIA) as shown in Figure 5. EIA is designed to obtain the spatial attention weight SA to direct the input image to focus more on the parts with information correlated to detection task. The formula SA is as follows:

$$SA = \sigma(W_{SA}^2 * \delta(W_{SA}^1 * (1 - EI) + b_{SA}^1) + b_{SA}^2), \quad (6)$$

where $EI \in \mathbb{R}^{1 \times H \times W}$ denotes the entropy image of the input image, $H$ and $W$ represent the height and width of the image. The term $(1-EI)$ adjusts the weights in the entropy image, assigning a higher weight to the portion containing information about the object in the original image and a lower weight to the remaining areas. $\sigma(\cdot)$, $\delta(\cdot)$ and $*$ denote the Sigmoid function, ReLU function and convolution operation respectively. $W$ and $b$ denote the weight parameters and bias of convolution. $SA \in \mathbb{R}^{1 \times H \times W}$ uses the Sigmoid function to map values to between 0 and 1. Then Adaptive Averaging Pooling is employed to plus a fully connected layer behind the feature extractor to return a quality score.

During the training stage, given two input pairs $(\tilde{x}_1, EI_{\tilde{x}_1})$ and $(\tilde{x}_2, EI_{\tilde{x}_2})$, along with two ground truth values $Q_1$ and $Q_2$, the output quality scores can be denoted by:

$$\begin{cases} q_1 = f\left(\tilde{x}_1; EI_{\tilde{x}_1}; \theta\right) \\ q_2 = f\left(\tilde{x}_2; EI_{\tilde{x}_2}; \theta\right) \end{cases}, \quad (7)$$

where $\theta$ refers to the network parameters. The whole loss function of DFIQA is as follows:

$$L_{DFIQA} = L1Loss(q_1, Q_1) + L1Loss(q_2, Q_2) + \lambda L(q_1; q_2), \quad (8)$$

where $\lambda$ is a hyperparameter, here set to 10, and $L(q_1; q_2)$ is margin-ranking loss, which is given below:

$$L(q_1, q_2) = \begin{cases} max(0, (q_2 - q_1) + \epsilon), & Q_1 > Q_2 \\ max(0, (q_1 - q_2) + \epsilon), & Q_1 < Q_2 \end{cases}, \quad (9)$$

where $\epsilon$ is a parameter, here set to 0.5.

In the testing phase, we directly selected one branch of DFIQA to predict the image quality.

### 3.3 Loss Function of EPL-UFLSID

To optimize the EPL-UFLSID effectively, we initially freeze weights of the feature extraction network at first 50 epochs, pre-trained on the COCO [16] dataset using the Resnet50 network. Then the network is unfreezed at last 50 epochs to optimize both the feature extractor and the back-stage network, which enables the feature extractor to fine-tune on the sonar dataset, enhancing the overall detection performance.

In the training stage, the loss function of EPL-UFLSID is defined as follows:

$$L_{EPL-UFLSID} = L_{cls} + L_{loc} + \lambda L_{QL} + L_{MSE}. \quad (10)$$

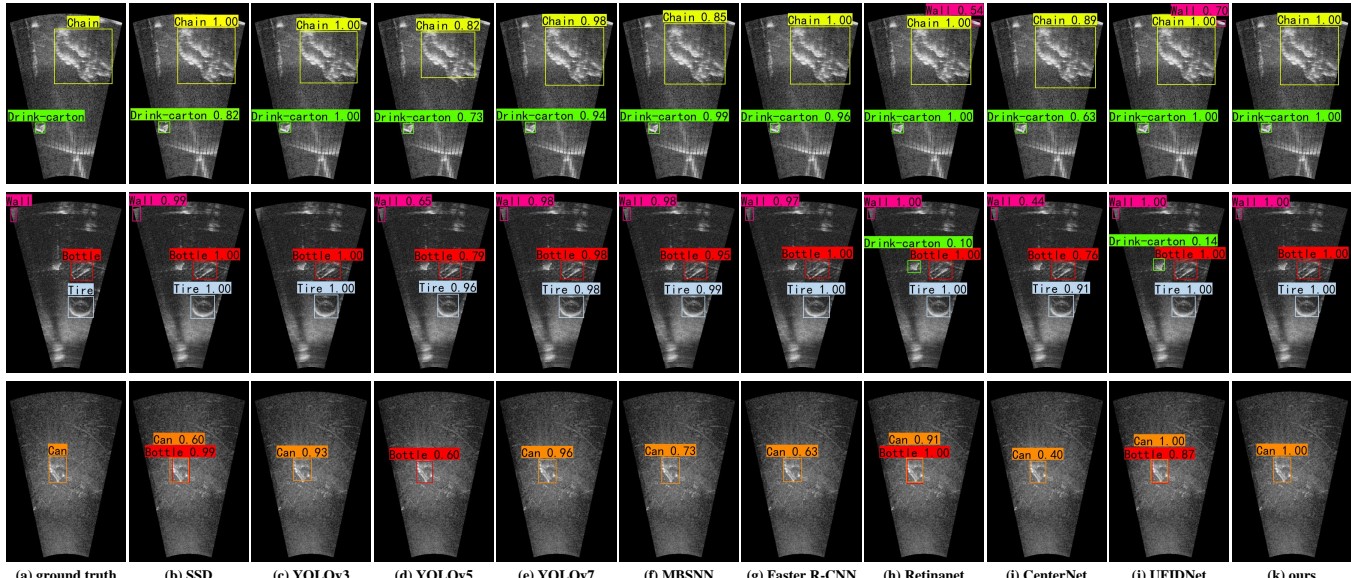

| (a) ground truth | (b) SSD | (c) YOLOv3 | (d) YOLOv5 | (e) YOLOv7 | (f) MBSNN | (g) Faster R-CNN | (h) Retinanet | (i) CenterNet | (j) UFIDNet | (k) ours |

**Figure 6: Qualitative comparison of the state-of-the-art object detection methods on the MDFD dataset. The third row of the resultant plots of (b), (h), and (j) algorithms contain two overlapping target boxes i.e., orange and red boxes, which is because the three algorithms incorrectly detect more targets. Besides, in the first and second row of reslutant plots, (h) and (j) algorithms all mistakenly detect the background as target.**

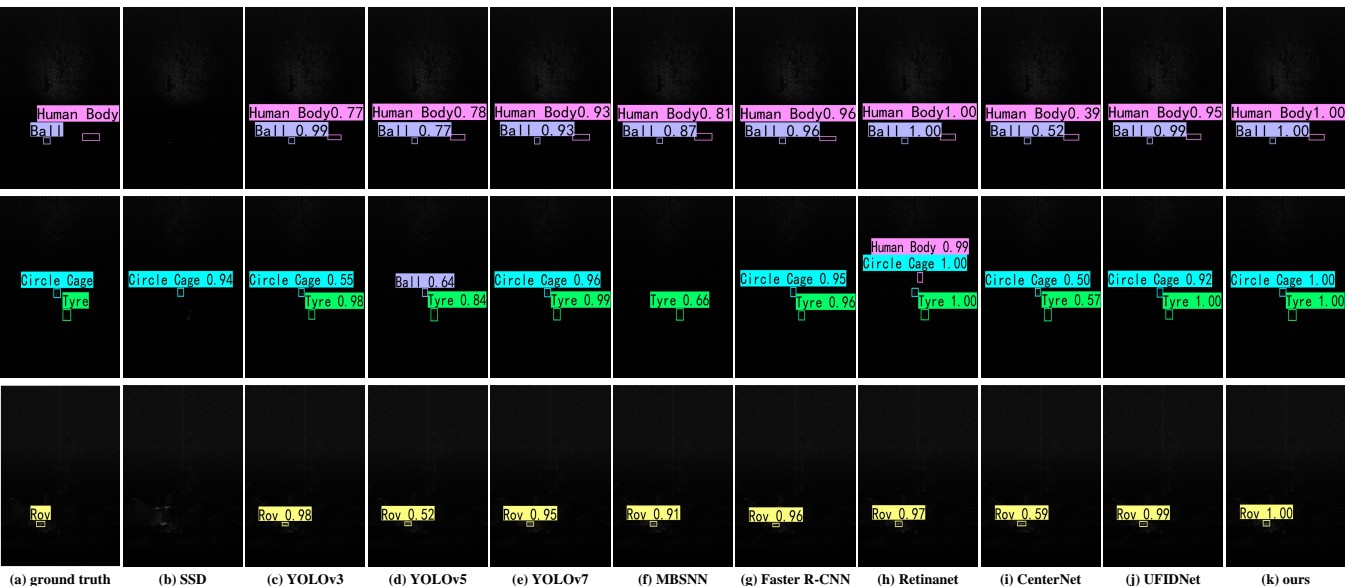

| (a) ground truth | (b) SSD | (c) YOLOv3 | (d) YOLOv5 | (e) YOLOv7 | (f) MBSNN | (g) Faster R-CNN | (h) Retinanet | (i) CenterNet | (j) UFIDNet | (k) ours |

**Figure 7: Qualitative comparison of the state-of-the-art object detection methods on the UATD dataset. Compared to the MDFD dataset, the objects of the UATD dataset are smaller and harder to detect. The second row of the resultant plots of (d) and (h) algorithms contain misdetections, where (d) incorrectly detects 'Circle Cage' as 'Ball' and (h) mistakenly detects the background as 'Human Body'. Besides, in the first and third row of resultant plots, none of the targets are detected by (b) algorithm.**

where $L_{cls}$ and $L_{loc}$ are Focal Loss [15] and Huber Loss, respectively. $\lambda$ represents a hyperparameter set to 0.003, and $L_{QL}$ denotes the quality loss, which is defined as:

$$L_{QL} = sigmoid(-Q(RI_i)), \qquad (11)$$

where $RI_i$ is the reconstructed image, $Q(RI_i)$ is the quality score of the reconstructed image. $L_{MSE}$ is L2-Loss employed to ensure the consistency between the $RI_i$ and the pseudo label $PL_i$ selected by

**Table 1: Quatitative comparison of the state-of-the-art object detection methods on the MDFD and UATD datasets. The best result in each column is in Red, and the second is in Blue. The baseline model of EPL-UFLSID is RetinaNet [15].**

| Method | Backbone | MDFD | | | | UATD | | | |
|---|---|---|---|---|---|---|---|---|---|
| | | $AP(\%)$ | $AP_{50}(\%)$ | $AP_{75}(\%)$ | $AR(\%)$ | $AP(\%)$ | $AP_{50}(\%)$ | $AP_{75}(\%)$ | $AR(\%)$ |
| SSD [17] | VGG-16 | 60.9 | 92.4 | 71.0 | 68.9 | 28.6 | 72.6 | 16.0 | 38.0 |
| YOLOv3 [21] | DarkNet-53 | 49.6 | 88.4 | 48.9 | 60.6 | 34.3 | 86.6 | 17.4 | 44.8 |
| YOLOv5 [8] | CSPDarkNet-53 | 44.9 | 76.6 | 48.4 | 60.8 | 36.6 | 86.3 | 20.4 | 46.9 |
| YOLOv7 [33] | CSPDarkNet-53 | 62.0 | 90.3 | 77.8 | 72.4 | 37.5 | 85.9 | 25.8 | 47.4 |
| MBSNN [34] | CSPDarkNet-53 | 54.1 | 83.2 | 62.6 | 65.8 | 34.7 | 81.4 | 19.1 | 44.3 |
| Faster R-CNN [23] | ResNet-50 | 58.2 | 94.2 | 64.8 | 66.5 | 20.3 | 60.6 | 6.7 | 41.0 |
| CenterNet [3] | ResNet-50 | 56.5 | 89.2 | 65.3 | 67.2 | 25.4 | 66.5 | 12.0 | 35.1 |
| RetinaNet [15] | ResNet-50 | 64.9 | 91.7 | 77.5 | 72.3 | 36.3 | 80.1 | 25.8 | 45.8 |
| UFIDNet [19] | ResNet-50 | 66.9 | 93.4 | 77.7 | 73.1 | 38.0 | 86.9 | 27.3 | 49.0 |
| **EPL-UFLSID(ours)** | ResNet-50 | 67.8 | 94.4 | 80.0 | 73.7 | 39.8 | 87.1 | 27.5 | 49.6 |

DFIQA, which is given as:

$$L_{MSE} = \frac{1}{|\Omega|} \sum_{\omega \in \Omega} (RI_i - PL_i)^2, \tag{12}$$

where $|\Omega|$ is the area of the map $\Omega$.

## 4 Experimental results

In this section, we first present various performance evaluation metrics. Then we compare our method with other sonar object detection methods. Finally, we perform some ablation experiments to validate the effectiveness of the proposed modules.

### 4.1 Datasets and Implementation Details

The effectiveness of the proposed method is demonstrated on the marine-debris-fls-dataset (MDFD) [26] and the underwater acoustic target detection (UATD) [36] dataset. The specific training dataset and testing set allocation and experimental implementation details will be in the supplementary materials.

### 4.2 Evaluation Metrics

#### 4.2.1 GMMDIP Evaluation Metrics.
Due to the absence of reference clean sonar images and our focus on generating pseudo labels optimized for detection, traditional metrics like PSNR and SSIM are not used for denoised sonar image quality assessment. Instead, we rely on objective metrics aligned with the EPL-UFLSID evaluation criteria to gauge GMMDIP network quality.

#### 4.2.2 DFIQA Evaluation Metrics.
Spearman Rank-Order Correlation Coefficient (SROCC) and Pearson Linear Correlation Coefficient (PLCC) are used to evaluate the performance of DFIQA. SROCC assesses the monotonicity of the method, while PLCC assesses the method's accuracy.

#### 4.2.3 EPL-UFLSID Evaluation Metrics.
For evaluating the performance of EPL-UFLSID, we use average precision (AP) and average recall (AR) in COCO [16]. Additionally, $AP_{50}$ and $AP_{75}$ are employed as metrics. $AP_{50}$ measures AP when the predicted bounding boxes overlap with the ground truth by at least 50%, while $AP_{75}$ measures it when the overlap is at least 75%.

**Table 2: DFIQA ablation comparison experiments on the MDFD and UATD datasets.**

| Model | L1 | EIA | MR | MDFD | | UATD | |
|---|---|---|---|---|---|---|---|
| | | | | SROCC | PLCC | SROCC | PLCC |
| DFIQA-MR-EIA | ✓ | | | 0.7677 | 0.6347 | 0.6054 | 0.5451 |
| DFIQA-MR | ✓ | ✓ | | 0.7736 | 0.6385 | 0.6149 | 0.5765 |
| DFIQA | ✓ | ✓ | ✓ | **0.7833** | **0.6439** | **0.6242** | **0.6162** |

### 4.3 Performance Comparison

In the section 4.2, we mention that the GMMDIP performance is represented by the evaluation metrics of EPL-UFLSID, and the performance of DFIQA is represented by SROCC and PLCC, which determines the EPL-UFLSID performance. Therefore we mainly describe the performance comparison of EPL-UFLSID and DFIQA with other methods.

#### 4.3.1 EPL-UFLSID Performance Comparison.
To demonstrate the performance of EPL-UFLSID, we compare its performance with 9 state-of-the-art detection methods on the MDFD and UATD datasets, respectively. The quantitative and qulitative results are illustrated in Tabel 1, Figure 6 and Figure 7.

From Table 1, Figure 6 and Figure 7, EPL-UFLSID stands out as significantly superior to the other 9 algorithms in terms of AP and AR results. EPL-UFLSID achieves an impressive AP of 67.8%, an AR of 73.7% on the MDFD dataset, which outperforms the second best algorithm by 0.9% and 0.6%, and an AP of 39.8%, an AR of 49.6% on the UATD dataset, which also outstands the second best algorithm by 1.8% and 1.6%. Moreover, our method outperforms in $AP_{50}$ and $AP_{75}$ results, demonstrating enhanced accuracy in target localization. In terms of $AR$, EPL-UFLSID consistently outperforms other methods, indicating its proficiency in identifying more objects compared to the competing methods. The superior performance of EPL-UFLSID benefits from GMMDIP and DFIQA module, which cooperatively generate efficient pseudo labels for detection model to extract cleaner features.

#### 4.3.2 DFIQA Performance Comparison.
The performance of DFIQA significantly influences the quality of

**Table 3: Performance comparison of EPL-UFLSID with pseudo labels generated by GMMDIP with different inputs at particular iteration on the MDFD and UATD datasets. The best result in each column is in Red, and the second is in Blue.**

| GMMDIP Input | iteration | MDFD | | UATD | |
|---|---|---|---|---|---|
| | | AP(%) | AR(%) | AP(%) | AR(%) |
| uniform_noise | 200 | 62.5 | 72.5 | 35.8 | 46.6 |
| | 300 | 65.5 | 73.2 | 36.9 | 46.3 |
| | 400 | 64.3 | 70.4 | 36.5 | 46.3 |
| | 500 | 64.6 | 72.1 | 36.8 | 46.6 |
| | 600 | 65.0 | 71.0 | 36.7 | 47.2 |
| GMM | 200 | 64.7 | 72.6 | 37.8 | 47.2 |
| | 300 | 67.2 | 73.7 | 37.4 | 48.2 |
| | 400 | 63.3 | 73.3 | 38.4 | 48.6 |
| | 500 | 63.1 | 73.6 | 37.3 | 47.2 |
| | 600 | 66.3 | 73.5 | 36.4 | 46.8 |

the selected pseudo labels, consequently impacting EPL-UFLSID detection performance. To assess the effectiveness of our method to select efficient pseudo labels, we conduct ablation experiments to analyze the performance of each module in DFIQA by SROCC and PLCC metrics.

The ablation experiments results of DFIQA on the MDFD and UATD datasets are illustrated in Table 2. Here the DFIQA network is divided into three sub parts: L1 Loss, EIA_Module, Margin_ranking Loss abbreviated as L1, EIA, MR, respectively. For the sake of space, the performance of DFIQA on the MDFD dataset will be mainly discussed. The baseline of DFIQA is 'DFIQA-MR-EIA' model, which only uses L1 for optimization, with a SROCC of 0.7677 and PLCC of 0.6347. 'DFIQA-MR' model additionally includes EIA with the help of entropy image, paying more attention to features beneficial for advanced tasks like detection, resulting in an improvment of 0.0059 SROCC and 0.0038 PLCC. 'DFIQA' model extends 'DFIQA-MR' model by adding MR, transforming the network into a siamese network and increasing sensitivity to ranking accuracy, hence SROCC and PLCC performance are further improved 0.0097 and 0.0054, achieving a SROCC of 0.7833 and PLCC of 0.6439. Besides, from the results in Table 2, DFIQA still has excellent performance on the UATD dataset.

## 4.4 Ablation study

### 4.4.1 Effectiveness of GMMDIP for EPL-UFLSID.

We compare the impact on EPL-UFLSID performance of denoised sonar images generated by two GMMDIP models, one of which uses GMM as input and the other uses uniform noise between 0 and 0.1 as input. Table 3 shows the results after introducing the denoised images obtained by GMMDIP with different inputs and after different iterations as pseudo labels into EPL-UFLSID for performance comparison on the MDFD and UATD datasets. We discover that the overall performance of EPL-UFLSID with GMMDIP using GMM as input achieves AP/AR values of 67.2%/73.7% and 38.4%/48.6% on the MDFD and UATD datasets, respectively, which is better than using uniform noise as input. In addition, the performance of using GMM

**Table 4: Comparison of EPL-UFLSID performance improvement by selected pseudo labels with different DFIQA models on the MDFD and UATD datasets.**

| DFIQA_model | MDFD | | UATD | |
|---|---|---|---|---|
| | AP(%) | AR(%) | AP(%) | AR(%) |
| DFIQA-MR-EIA | 65.7 | 72.5 | 37.0 | 46.6 |
| DFIQA-MR | 66.7 | 72.7 | 37.6 | 47.3 |
| DFIQA | 67.3 | 73.1 | 39.1 | 48.8 |

**Table 5: Performance comparison of EPL-UFLSID and EPL-UFLSID with $L_{QL}$ on the MDFD and UATD datasets..**

| model | MDFD | | UATD | |
|---|---|---|---|---|
| | AP(%) | AR(%) | AP(%) | AR(%) |
| Baseline (Retinanet[15]) | 64.9 | 72.3 | 36.3 | 45.8 |
| EPL-UFLSID | 67.3 | 73.1 | 39.1 | 48.8 |
| EPL-UFLSID+$L_{QL}$ | 67.8 | 73.7 | 39.8 | 49.6 |

as input is superior compared to using uniform noise as input on the whole.

### 4.4.2 Effectiveness of DFIQA for EPL-UFLSID.

DFIQA is designed to select the most favorable images for detection from a batch of denoised images of different qualities generated by GMMDIP, which also avoids manually picking the best detection-friendly denoised images as pseudo labels. As shown in Table 4, three DFIQA models mentioned in Table 2 have different performance gains for EPL-UFLSID. As SROCC and PLCC of DFIQA improves, EPL-UFLSID achieves improvement up to 1.6%/0.6% and 2.1%/2.2% AP/AR on the MDFD and UATD datasets, respectively. Moreover, to further demonstrate the effectiveness of DFIQA, we use the pre-trained DFIQA as additional supervision to optimize EPL-UFLSID, resulting in the model EPL-UFLSID+$L_{QL}$. The results, shown in Table 5, indicate that the AP/AR values on the MDFD and UATD datasets improve by 2.9%/1.4% and 3.5%/3.8%, respectively, compared to the baseline model. Additionally, compared to the EPL-UFLSID model without additional supervision $L_{QL}$, the improvements are 0.5%/0.6% and 0.7%/0.8%, respectively. These results fully verify the effectiveness of DFIQA for EPL-UFLSID.

## 5 Conclusion

In this paper, we propose EPL-UFLSID, a novel approach for enhancing object detection in underwater forward-looking sonar images. Our method introduces efficient pseudo labels for cleaner feature extraction, relying on two key modules: GMMDIP and DFIQA. GMMDIP generates denoised sonar images of different qualities without reference clean images, while DFIQA filters the top scores denoised images as detection-friendly pseudo labels. The collaboration between GMMDIP and DFIQA enables EPL-UFLSID to extract cleaner features from efficient pseudo labels, thus improving detection performance. Extensive experiments on the MDFD and UATD datasets validate the effectiveness of EPL-UFLSID.

# Acknowledgments

This work was supported in part by the National Natural Science Foundation of China under Grant 61931022 and Grant 62271301; in part by the Shanghai Science and Technology Program under Grant 22511105200; in part by the Shanghai Excellent Academic Leaders Program under Grant 23XD1401400; and in part by the Natural Science Foundation of Shandong Province under Grant ZR2022ZD38.

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
