# OpenReview forum: "EPL-UFLSID: Efficient Pseudo Labels-Driven Underwater Forward-Looking Sonar Images Object Detection"
_acmmm.org/ACMMM/2024/Conference — MM2024 Poster_

### Official Review · Reviewer_TbXc · 2024-05-14

**Rating:** 3
**Confidence:** 2

**Summary:**

This work points out that it is unreasonable for the view denoised image to be used as a pseudo-label to assist the network to extract clean features, because there may be noise in the denoised sonar image that cannot be observed by the human eye, thus reducing the performance of the subsequent detection model. Based on this, an efficient pseudo-label-driven target detection algorithm for underwater forward-looking sonar images (EPL-UFLSID) was proposed. The algorithm relies on two key innovative modules: GMMDIP, an unsupervised sonar image denoising network; and DFIQA, a detection-friendly image quality assessment network. The former is used to generate high-quality denoised sonar images without reference to clean images. The latter is used to select the most effective pseudo-labels for detection. Comparative experiments and ablation experiments on MDFD and UATD datasets verify the effectiveness of EPL-UFLSID.

**Strengths:**

This work paid attention to the impact of view denoising and detection denoising on detection performance, and proposed a targeted pseudo-label image generation method from the perspective of detection performance. Combining the Gaussian mixture model and depth image prior related content, a Gaussian mixture model based Deep Image Prior network (GMMDIP) is proposed in the field of sonar target detection to generate a series of reliable denoised sonar images as pseudo labels. And a twin-like network DFIQA for image quality assessment is proposed to select the most effective pseudo-labels for detection.

**Limitations:**

1. The algorithm as a whole is just a simple combination of existing technical modules. It does not propose special structures, concepts and technologies, and the overall innovation is insufficient.
2. The process of selecting the optimal image for pseudo label detection from a series of denoised images with different denoising qualities generated by GMMDIP is not described clearly enough.
3. How is the result of the input image in Figure 1 after passing through the target detection backbone network related to the pseudo-labeled image?
4. The sonar images of the UATD data set are subjectively not clear enough (too dark). Although this reflects the charm of deep learning in target detection, the presentation effect is confusing.
5. There are indeed relatively few public data sets for sonar, but we still hope to use more data sets to conduct comparative experiments to prove the generalization performance of the algorithm.
6. In the ablation experiment, the performance improvement of the proposed module is not significant; and in the comparison experiment with other algorithms, the performance improvement is also not significant. The effectiveness of the algorithm needs to be further verified.

**Suitability:**

3

---

### Official Review · Reviewer_V3BX · 2024-05-17

**Rating:** 3
**Confidence:** 3

**Summary:**

This paper introduces the Efficient Pseudo Labels-Driven Underwater Forward-Looking Sonar Images Object Detection algorithm (EPL-UFLSID), which leverages denoised sonar images as pseudo labels to enhance object detection in challenging underwater environments. The algorithm incorporates a Gaussian Mixture Model based Deep Image Prior (GMMDIP) to generate clean sonar images and utilizes a Detection-Friendly Image Quality Assessment network (DFIQA) to filter efficient pseudo labels for improved detection performance. EPL-UFLSID aims to extract cleaner features from pseudo labels to enhance object detection accuracy in underwater sonar images.

**Strengths:**

- EPL-UFLSID optimizes the detection process in underwater sonar images by integrating GMMDIP and DFIQA, and has undergone rigorous evaluation and comprehensive ablation experiments on the MDFD and UATD datasets to ensure technical accuracy.
- The paper provides clear explanations of the algorithm's components, including GMMDIP and DFIQA, making it accessible to readers interested in underwater object detection.

**Limitations:**

- Although the method of using pseudo-labels to guide the target detection model in EPL-UFLSID is effective, similar approaches have been explored in previous studies. This paper mainly enhances performance by improving the accuracy of pseudo-labels through designed modules, which is somewhat lacking in terms of innovation.
- The paper lacks a detailed discussion on the computational efficiency of EPL-UFLSID, such as the computational resources required for denoising sonar images using GMMDIP and selecting pseudo-labels using DFIQA. Computational efficiency in underwater sonar scenarios has a significant impact on practical applications, making such a discussion necessary.

**Suitability:**

3

---

### Official Review · Reviewer_difh · 2024-05-20

**Rating:** 4
**Confidence:** 3

**Summary:**

This paper introduces an efficient pseudo label-driven underwater forward-looking sonar images object detection algorithm (EPL-UFLSID). The algorithm employs a Gaussian Mixture Model-based Deep Image Prior (GMMDIP) to generate denoised sonar images and uses a Detection-Friendly Image Quality Assessment network (DFIQA) to select efficient pseudo labels. This process helps to extract cleaner features and enhance detection performance. Experimental results demonstrate that EPL-UFLSID outperforms state-of-the-art underwater sonar image object detection algorithms on two datasets.

However, I still have some concerns (see the Limitations section for details).

**Strengths:**

A novel algorithm named EPL-UFLSID is proposed, designed to enhance sonar image object detection performance by introducing efficient pseudo labels. These pseudo labels guide the detection backbone to extract cleaner, more detection-friendly features.

An unsupervised sonar image denoising network, GMMDIP, is introduced. This network generates high-quality denoised sonar images, which serve as pseudo labels to further improve detection performance.

A detection-friendly image quality assessment network named DFIQA is presented, which is tasked with selecting the most effective pseudo labels for detection purposes.

**Limitations:**

1. The paper indicates that the authors use reconstructed images generated by GMMDIP as pseudo labels. What, then, is the specific role of the Reconstructed Module depicted in Figure 2?

2. Recently, there have been numerous studies on the application of GMM in computer vision, such as "Learned Image Compression with Discretized Gaussian Mixture Likelihoods and Attention Modules." Despite this, GMM does not show particularly superior performance in Table 3.

3. Are the iterations based on different noise levels in Table 3 the same or different?

4. Tables 4 and 5 demonstrate the impact of DFIQA on model performance. However, the quantitative results indicate that the improvements are not substantial. For example, in Table 4, DFIQA based on the MDFD dataset improves the AR metric by only 0.1% compared to DFIAQA-MR.

5. While this work is very interesting, the overall model described seems quite complex. Could you provide a comparative analysis of parameters, FLOPs, and other complexity metrics?
The authors do not seem to have validated the effectiveness of the designed loss function. It would be beneficial to include an ablation study on the loss function.

6. Additionally, some minor details need correction. For example, on page 2, line 123, the abbreviation "DIP" is not defined upon its first use and is only explained on page 3, line 280. Similar issues throughout the paper should be addressed.

**Suitability:**

2

---

### Official Review · Reviewer_eS5K · 2024-05-25

**Rating:** 4
**Confidence:** 3

**Summary:**

This work first proposes an image denoising network GMMDIP to generate a series of denoised sonar images for each raw sonar image, then an image quality assessment network DFIQA is proposed to select the most detection friendly image from the generated denoised sonar images, the most detection friendly image is added as extra supervision information to train the detection network to learn better/clean features.

**Strengths:**

The proposed method presents some novelty

**Limitations:**

The logic of the proposed EPL-UFLSID doesn’t sounds reasonable in current statement. Some important logics should be explained as follows:

1. Since previous works show that the denoised sonar images may not necessarily improve object detection, please explain why some denoised images/pseudo labels generated by GMMDIP can boost the detection performance rather than degrade the performance? Why GMMDIP can generate detection-friendly images?

2 Please explain why higher MOS value indicates better detection-friendly? What datasets you used to train the YOLOv3, SSD and RetinaNet?

3. In DFIQA, please explain what the entropy images is and why the entropy images are used as one of the inputs.

**Suitability:**

2

---

### Meta-Review · Area_Chair_2dgS · 2024-06-30

**Recommendation:** Accept (Poster)
**Confidence:** 3

**Metareview:**

This paper received contrasting reviews, before and after rebuttal, that is,  2 BA and 2 BR.

Main concerns regard weak novelty and justification of the proposed approach, the need of more details/explanation in some parts of the pipeline, request of additional comments on the experimental results, and quoting the computational cost.
The rebuttal reasonably responds to such issues in a more or less satisfactory way for the positive reviewers, but not for the other, more critical ones.

The AC read the comments and rebuttal, and looked at the paper too: in the end, even if he recognizes some critical aspects of the work, he is overall agreeing more to the former scholars. For this reasons, this paper is considered acceptable for publication to ACM MM 24.